# The Anti-Microbial Peptide Citrocin Controls *Pseudomonas aeruginosa* Biofilms by Breaking Down Extracellular Polysaccharide

**DOI:** 10.3390/ijms25074122

**Published:** 2024-04-08

**Authors:** Liyao Wang, Jiaqi Zheng, Wenchao Hou, Chaowen Zhang, Jie Zhang, Xuanbo Fan, Hongliang Zhang, Yuzhu Han

**Affiliations:** 1College of Animal Science and Technology, Southwest University, Chongqing 402460, China; 220233940@seu.edu.cn (L.W.); jiaqiii.zheng@connect.polyu.hk (J.Z.); houwenchao@email.swu.edu.cn (W.H.); 12316010@zju.edu.cn (C.Z.); zhangjiedk@swu.edu.cn (J.Z.); fan649574309@email.swu.edu.cn (X.F.); sy20233040871@cau.com (H.Z.); 2College of Life Science and Technology, Southeast University, Nanjing 211189, China; 3Department of Food Science and Nutrition, The Hong Kong Polytechnic University, Hong Kong SAR, China; 4College of Animal Science and Technology, China Agricultural University, Beijing 100083, China; 5Chongqing Key Laboratory of Herbivore Science, Chongqing 402460, China

**Keywords:** *Pseudomonas aeruginosa*, extracellular polysaccharide, biofilm, swarming motility, metabolomics

## Abstract

Citrocin is an anti-microbial peptide that holds great potential in animal feed. This study evaluates the anti-microbial and anti-biofilm properties of Citrocin and explores the mechanism of action of Citrocin on the biofilm of *P. aeruginosa*. The results showed that Citrocin had a significant inhibitory effect on the growth of *P. aeruginosa* with a minimum inhibitory concentration (MIC) and minimum bactericidal concentration (MBC) of 0.3 mg/mL. All five concentrations (1/4MIC, 1/2MIC, MIC, 2MIC, and 4MIC) of Citrocin inhibited *P. aeruginosa* biofilm formation. Citrocin at the MIC, 2MIC and 4MIC removed 42.7%, 76.0% and 83.2% of mature biofilms, respectively, and suppressed the swarming motility, biofilm metabolic activity and extracellular polysaccharide production of *P. aeruginosa*. Metabolomics analysis indicated that 0.3 mg/mL of Citrocin up- regulated 26 and down-regulated 83 metabolites, mainly comprising amino acids, fatty acids, organic acids and sugars. Glucose and amino acid metabolic pathways, including starch and sucrose metabolism as well as arginine and proline metabolism, were highly enriched by Citrocin. In summary, our research reveals the anti-biofilm mechanism of Citrocin at the metabolic level, which provides theoretical support for the development of novel anti-biofilm strategies for combatting *P. aeruginosa*.

## 1. Introduction

*Pseudomonas aeruginosa* is a Gram-negative rod-shaped bacterium that is widely found in water, soil, plants and animals [1]. It is an opportunistic pathogen that tends to colonize immunocompromised patients [2]. *P. aeruginosa* has a prevalence of 10–20% in hospitals [3] and can cause a variety of chronic and acute illnesses such as pneumonia, urinary tract infections and bacteremia [4]. Furthermore, this pathogen exhibits high drug resistance due to its strong biofilm-forming ability, further complicating the treatment of its infections [5]. Biofilms are known to be a bacterial cell-coordinated community structure encapsulated in self-produced extracellular polymers (EPS) composed of extracellular polysaccharide, proteins, extracellular DNA (eDNA) and lipids [6,7].

The primary building block of EPS is extracellular polysaccharide [3]. *P. aeruginosa* can synthesize at least three types of polysaccharides including Psl, Pel and Alginate, each of which provides unique physiological characteristics to the matrix [1]. Among them, Psl consists of repeats of L-rhamnose, D-glucose and D-mannose, which are mainly found on the cell surface and maintain biofilm structure by facilitating cell–cell surface interactions [8]. In addition, Psl enhances matrix elasticity and cross-linking, favoring the establishment of biofilm microcolonies [9]. Pel is a cationic extracellular polysaccharide that enhances biofilm specific resistance to antibiotics and improves the stability of the bioepidermis [10,11,12]. Alginate is a linear anionic polysaccharide containing α-L-guluronic acid with β-1-4 glycosidic linkages and β-D-mannuronate [12], which can protect biofilm cells from adversarial environments and improve surface adhesion [3,13].

The matrix proteins of *P. aeruginosa* are mostly composed of flagella and type IV pili (cell-surface appendages). The flagellum is made up of flagellin, which is necessary for bacterial cell adhesion, swimming motility and biofilm development [14]. Type IV pili are protein fibers produced on the surface of the bacterial cell and are involved in colonization, biofilm formation and bacterial translocation during infection [15].

EPS shields biofilms from removal by the host immune system and external adverse environmental influences [16]. In addition, the capability of EPS to limit the penetration of anti-microbial agents allows the minimum inhibitory concentration (MIC) of anti-microbial agents on *P. aeruginosa* biofilms to be 100–1000 times higher than that required to treat bacterioplankton [17,18]. Biofilms are one of the main factors responsible for the development of resistance to antibiotics in *P. aeruginosa* [19,20]. Currently, various bacteriostatic agents such as anti-microbial peptides and phages, as well as the related mechanisms, have been extensively studied. However, the elucidation of pathogenic mechanisms of *P. aeruginosa* is still lacking because of the complexity and variability of biofilm [21]. Therefore, further studies on anti-microbial–biofilm interactions play a crucial role in finding new drug targets and designing more effective anti-biofilm strategies.

Citrocin is an anti-microbial peptide that is widely used in animal feeds, and it is added to different animal feeds at concentrations of 0.1–0.5 g/t. Citrocin is an active peptide produced by *Saccharomyces cerevisiae* Sa-10. It consists of 19 amino acid residues (GGVGKIIEYFIGGGVGRYG), with a molecular weight of 1.9 kDa. Citrocin is heat-resistant and not easily hydrolyzed by chemical and proteolytic enzymes, ensuring its safety in use. Previous studies have shown that the anti-microbial peptide Citrocin can inhibit the growth of *E. coli* [22]. However, the anti-microbial and anti-biofilm efficacy and mechanism of Citrocin against *P. aeruginosa* have not yet been reported.

In this study, we evaluated the anti-bacterial and anti-biofilm effects of Citrocin on the opportunistic pathogen *P. aeruginosa* and discussed the effects of Citrocin on *P. aeruginosa* biofilm in terms of biofilm structure, biofilm metabolic activity, bacterial swarming motility, extracellular polysaccharide production and metabolomics. 

## 2. Results 

### 2.1. Evaluation of Inhibitory Effect

The anti-microbial peptide Citrocin had a significant inhibitory effect on the growth of *P. aeruginosa* (Figure 1A). Oxford cups (b and c) spiked with Citrocin had clear translucent circles and no discernible bacterial growth when compared with the control group (a). Furthermore, there was no obvious difference in the inhibition circles between the heat-treated group (c) and the non-heat-treated group (b), indicating that the anti-microbial peptide Citrocin is heat-stable and would not be affected by a high temperature of 121 °C.

### 2.2. Minimum Inhibitory Concentration (MIC) and Minimum Bactericidal Concentration (MBC)

Four test tubes (a, b, c, and d) were turbid with visible bacterial growth, and two test tubes (e and f) had clear liquid and no visible bacterial growth (Figure 1B), which means the MIC of the anti-microbial peptide Citrocin was 0.3 mg/mL (based on the judgment standard in Section 4.2.2). Subsequently, the bacterial liquids in the six test tubes were sequentially inoculated on LB agar plates, and the results showed no colony growth after the bacterial solution was treated with 0.3 mg/mL of Citrocin (Figure 1C). Therefore, the MBC of the anti-microbial peptide Citrocin against *P. aeruginosa* was 0.3 mg/mL.

### 2.3. Biofilm Inhibition Assay

The effect of different concentrations of Citrocin solution on *P. aeruginosa* biofilm formation was determined based on the crystal violet staining (CV) method [23]. As shown in Figure 2A, 1/4MIC of Citrocin exhibited a significant inhibitory effect (*p* < 0.05) on the biofilm development of *P. aeruginosa*. When the concentration was greater than or equal to 1/2MIC, the anti-microbial peptide Citrocin showed highly significant (*p* < 0.01) biofilm inhibition of *P. aeruginosa*. As the concentration of Citrocin increased, the amount of biofilm formation gradually decreased, indicating a negative correlation between the amount of *P. aeruginosa* biofilm production and the concentration of the anti-microbial peptide Citrocin.

### 2.4. Biofilm Structure Detection

*P. aeruginosa* without Citrocin treatment generated dense biofilms (Figure 2C). Compared with the control group, the biofilm under 1/4MIC treatment became obviously looser and sparser, and the production of biofilm decreased as the concentration of the anti-microbial peptide Citrocin increased. When the concentration of the anti-microbial peptide reached 2MIC and 4MIC, almost no biofilm was developed, which reveals that the anti-microbial peptide Citrocin could significantly inhibit biofilm formation in *P. aeruginosa.*

### 2.5. Measurement of Biofilm Biomass

Citrocin at a concentration of 1/4MIC and 1/2MIC displayed an apparently stimulating impact rather than the clearing effect on the biofilm that was already established for *P. aeruginosa* (Figure 2B), but the difference was not statistically significant (*p* > 0.05). In contrast, Citrocin at MIC, 2MIC, and 4MIC cleared 42.7%, 76.0%, and 83.2% of mature biofilms, respectively. This suggests that the effect of typical concentrations of anti-bacterial agents on the biofilm is not evident once the biofilm has developed, and higher doses of medications are required to remove the biofilm.

### 2.6. Biofilm Metabolic Activity Assay 

As shown in Figure 3A, the anti-microbial peptide Citrocin at sub-MICs (1/4MIC and 1/2MIC) stimulated biofilm metabolic activity, whereas the MIC, 2MIC, and 4MIC of Citrocin were able to inhibit the biofilm metabolic activity of *P. aeruginosa* extremely significantly (*p* < 0.01). This remains in good agreement with the findings from 2.5.

### 2.7. Swarming Motility Assay

The swarming motility of bacteria is essential for biofilm formation and can promote *P. aeruginosa* to attach to contact surfaces or invade other parts of the host, aggravating bacterial infections [24]. The effect of different concentrations of Citrocin on the motility vigor of *P. aeruginosa* was detected based on the semi-solid agar plate method. Figure 3C illustrates that there was basically no difference in colony diameter between the 1/4MIC and 1/2MIC treatment groups compared with the control group, indicating that Citrocin of 1/4MIC and 1/2MIC could not affect the motility of *P. aeruginosa*. However, MIC, 2MIC and 4MIC treatment groups all evidently suppressed the motility of *P. aeruginosa* (*p* < 0.01) and showed a certain dose dependence (Figure 3D).

### 2.8. Determination of Extracellular Polysaccharide Content

During biofilm formation, bacterial cells continuously secrete polysaccharides, proteins and other substances externally to form a complex and strong biofilm shell to resist external drug interference [25]. In this study, the phenol-sulfuric acid method was used to assess the effect of different concentrations of Citrocin solution on extracellular polysaccharide production in *P. aeruginosa* [26]. The sub-MICs of the anti-microbial peptide Citrocin could not reduce the production of extracellular polysaccharide by *P. aeruginosa* (Figure 3B). Citrocin with MICs could significantly inhibit extracellular polysaccharide production (*p* < 0.05), while both 2MIC and 4MIC concentrations of Citrocin drastically cut down the biofilm extracellular polysaccharide (*p* < 0.01).

### 2.9. Metabolomics Analysis

#### 2.9.1. Effect of Different Treatments on the Metabolic Profile of Biofilm

A principal component analysis of the intracellular metabolites of the nine samples is shown in Figure 4A, with three biological replicates within each group showing no outliers, indicating the stability of the experimental procedure. Samples from the control (CK) and Citrocin-treated groups (LP and HP), which represent variations in metabolic profiles across samples, demonstrated a distinct tendency toward separation from the low-peptide-treated (LP) and high-peptide-treated (HP) groups. Metabolomics data from three treatments were compared using OPLS-DA to realize optimal within-group separation, and a clear separation was obtained from different treatments (Figure 4B). The OPLS-DA model was verified using alignment tests. The results of 200 permutation tests are shown in Figure 4C, and the intercept indicates that the model was reliable and not overfitted. The value of Q2 of the OPLS-DA model was generally > 0.5, and the value of R2 was close to 1, which reveals that the model was stable and suitable for prediction, and could be used to identify the differences between groups. In total, 351 metabolites were detected after treatment with Citrocin solution. Of these, 34 metabolites were up-regulated and 53 metabolites were down-regulated in CK vs. LP, and 26 metabolites were up-regulated and 83 metabolites were down-regulated in CK vs. HP, suggesting that the addition of the anti-microbial peptide Citrocin significantly affects biofilm metabolism (Figure 4D).

#### 2.9.2. Identification of Differential Metabolites

In total, 351 differential peaks were detected with a simultaneously satisfying *p*-value < 0.05 and VIP (OPLS-DA) > 1 as screening conditions. After comparison, 45 and 58 differential metabolites were identified in the CK vs. LP and CK vs. HP groups, respectively. The heatmap (Figure 5) shows that the control (CK) and Citrocin-treated groups (LP and HP) were clearly separated and the differential metabolites changed with increasing concentrations of added peptide solution. The metabolites can be categorized into different classes (Figure 6A), and the majority were acids and their derivatives (16% and 26% in the CK vs. LP and CK vs. HP groups, respectively), fatty acids (18% and 16% in the CK vs. LP and CK vs. HP groups), organic acids and their derivatives (16% and 11% in the CK vs. LP and CK vs. HP groups), and sugars (13% in both CK vs. LP and CK vs. HP groups). (Although both amino acids and fatty acids are organic acids, there were many types and amounts of amino acids and fatty acids affected in this study, so these two acids are listed separately). Amino acids, fatty acids and sugars are all major constituents of the extracellular matrix of biological membranes, suggesting that the anti-microbial peptide Citrocin significantly altered the composition and content of the biofilm extracellular matrix. In addition, the composition and level of fatty acids were closely related to the fluidity of the bacterial cell membrane, through which the cells responded to external stress. Our results revealed that 71.43% and 66.67% of fatty acids were down-regulated in the CK vs. LP group and the CK vs. HP group, respectively, which indicates that the anti-microbial peptide Citrocin has a certain effect on the fluidity of the cell membrane of *P. aeruginosa*.

#### 2.9.3. Metabolic Pathway Analysis

The KEGG database was used to conduct a pathway enrichment analysis of the differential metabolites (Figure 6B). Pathways reported to have impact values higher than 0.1 were selected for further analysis. In the CK vs. LP group, metabolic pathways with impact values higher than 0.1 included arginine and proline metabolism, starch and sucrose metabolism, and glutathione metabolism. In the CK vs. HP group, metabolic pathways with effect values higher than 0.1 included arginine and proline metabolism, arginine biosynthesis, starch and sucrose metabolism, D-Glutamine and D-glutamate metabolism, and alanine, aspartate and glutamate metabolism. A metabolic network was constructed based on the metabolic pathways with relatively significant differences (Figure 7), demonstrating that enhanced glycolysis and the metabolism of multiple amino acids resulted in the aberrant activation of the TCA cycle. This may be related to the stress of the bacteria altering their metabolic pathways to adapt to the hostile environment after Citrocin treatment.

## 3. Discussion 

It has been reported that 65% of microbial diseases and more than 80% of chronic infections are associated with bacterial biofilms [27]. EPS secreted by biofilm is able to limit bacterial spread into other unfavorable environments, such as those with UV light, an extreme pH and desiccation. It also strengthens the biofilm structure, making the bacteria more resistant to anti-microbial agents, which results in treatment failures and recurrent infections [28].

In this study, the anti-biofilm properties of Citrocin against *P. aeruginosa* were tested at five concentrations, including two low concentrations (1/4MIC and 1/2MIC) and three high concentrations (MIC, 2MIC and 4MIC). Notably, *P. aeruginosa* biofilm production was effectively inhibited by the sub-minimum inhibitory concentrations (sub-MICs) of Citrocin; however, existing mature biofilms remained unaffected. This implies that Citrocin can effectively suppress the growth of *P. aeruginosa*, thus preventing the development of biofilm. Once the biofilm was formed, a higher dosage of the drug was required to achieve clearance. Our finding is consistent with previous studies reporting that biofilms are generally difficult to remove after formation and that anti-microbial treatments are less effective against biofilms than their equivalent planktonic counterparts [29]. Meanwhile, the current investigation demonstrated that Citrocin at sub-MICs stimulated *P. aeruginosa* to form more biofilms rather than eliminating the existing ones. Previous research [28,30] provides strong support for this conclusion since bacteriostatic agents can alter the phenotypes of bacteria at low concentrations, which in turn promotes the production of biofilms. Therefore, in order to prevent the administration of inappropriate therapies in the context of biofilm infections, it is crucial to understand the clinical impact of anti-microbial drugs on biofilm development.

The early stage of biofilm formation is the migration, adhesion and colonization of planktonic bacteria [31]. *P. aeruginosa* primarily employs flagellum-driven swarming motility in this phase to reach available conditioned surfaces and undergo initial adhesion [32]. Type IV pili or other surface adhesins then promote attachment, making surface bonding irreversible and completing the initial colonization. In this investigation, we discovered that the anti-microbial peptide Citrocin was capable of restraining the movement of *P. aeruginosa* (Figure 3C). Meanwhile, metabolomics data showed that elevated levels of amino acids associated with protein synthesis (arginine, proline, leucine, glutamate, etc.) led to accelerated related metabolism (Figure 7). Additionally, *P. aeruginosa* biofilm matrix proteins including the flagellum and type IV pili are intimately involved in swimming motility. Citrocin, an anti-microbial peptide, is therefore thought to break down type IV pili proteins and flagellar proteins by speeding up the metabolism of amino acids. This causes *P. aeruginosa* to lose its ability to migrate and adhere, which lowers the initial colonization rate of the bacteria on the contact surface. The flagellum and type IV pili, which are closely related to swimming behavior, are both matrix proteins of *P. aeruginosa* biofilm. Therefore, it is hypothesized that the anti-microbial peptide Citrocin breaks down flagellar proteins and type IV pili proteins by accelerating the metabolism of amino acids, which results in the loss of migration and adhesion ability of *P. aeruginosa*, and thus reduces the initial colonization rate of the bacteria on the contact surface.

Upon the completion of colonization on the contact surface, the bacteria multiply and exocytose EPS wrap around microcolonies, thereby increasing cellular resistance to stress (antibiotics, osmosis and oxidation) [33]. Therefore, any disruption of EPS synthesis will inevitably affect the biofilm structure, decreasing its resistance and exposing the bacteria to damage from antibiotics and drugs [34]. Citrocin solution had a decreasing effect on the extracellular polysaccharide content of *P. aeruginosa* (Figure 3B). The comparison of biofilm metabolites before and after treatment with Citrocin solution revealed that trehalose, maltose, lactose, gluconic acid and other saccharides were significantly up-regulated, resulting in the related starch and sucrose metabolism and galactose metabolism, and the content of fatty acids was highly down-regulated. This suggests that the energy generated from sugar metabolism does not accumulate but is fully consumed. Therefore, the present study hypothesized that the anti-microbial peptide Citrocin utilizes sugars by means of accelerated energy metabolism, leading to the burst of extracellular polymer shells and altering the structure of biofilms. At the same time, the rupture of the EPS facilitates the entry of bacteriostatic substances, which will be more advantageous to the treatment of biofilm.

## 4. Materials and Methods

### 4.1. Strain and Culture

*P. aeruginosa* ATCC 15442 used in this study was obtained from the Grassland Microbiology Laboratory of Southwest University. The anti-microbial peptide Citrocin was supplied by Beijing Enhalor International Tech Co., Ltd. (Beijing, China). LB broth (Beijing Luqiao, China) was used for bacterial incubation and biofilm formation. The bacteria stored in the glycerol tubes were inoculated on LB agar plates and incubated at 37 °C for 18 h. Subsequently, single colonies grown on the plates were inoculated into fresh LB broth, and the bacterial suspensions were cultured in a shaker at 150 rpm for 12 h.

### 4.2. Anti-Microbial Activity

#### 4.2.1. Agar Diffusion Test

*P. aeruginosa* bacterial suspension (1 × 10^7^ CFU/mL) was evenly spread on the LB agar plate. Oxford cups filled with 200 μL of Citrocin solution (final concentration of 0.3 mg/mL) were placed on the LB agar plate. The same amount of PBS was used as a blank control. All the plates were incubated at 37 °C for 24 h. After incubation, the zones of inhibition around the Oxford cups were measured and recorded [34,35,36].

#### 4.2.2. Minimum Inhibitory Concentration (MIC) and Minimum Bactericidal Concentration (MBC)

The broth dilution method combined with the plate transfer method was used to determine the minimum inhibitory concentration (MIC) and minimum bactericidal concentration (MBC) of the anti-microbial peptide [37]. Six autoclaved and numbered test tubes were prepared and dispensed sequentially with 2 mL of LB medium, Citrocin solution was added to the first test tube, 2 mL of the mixture was transferred to the next test tube, and so on, while 2 mL of the mixture was obtained from the fifth test tube and discarded. The final concentrations of the anti-microbial peptides were 0.3, 0.15, 0.075, 0.0375 and 0.01875 mg/mL, respectively. PBS solution was taken as a blank control. Then, 100 μL of *P. aeruginosa* suspension (10^7^ CFU/mL) was added into 6 test tubes and incubated at 37 °C for 18 h. The MIC against *P. aeruginosa* was evaluated using the turbidimetric method. The lowest Citrocin concentration with no visible growth in the tubes was defined as the MIC. The MBC was determined by inoculating each of the six tubes with LB agar plates using the plate transfer method and observing the growth of the colonies after 48 h of incubation.

### 4.3. Biofilm Inhibition Test

#### 4.3.1. Biofilm Biomass Assay

The *P. aeruginosa* bacterial suspension (10^7^ CFU/mL) was transferred to a flat-bottomed 96-well plate [38], and Citrocin solution was added to obtain final concentrations of 1/4MIC, 1/2MIC, MIC, 2MIC and 4MIC. Wells without the anti-microbial peptide treatment were used as a negative control, and cultures not inoculated with bacterial solution were regarded as blank controls. The 96-well plates were incubated at 37 °C for 72 h in order to allow the formation of mature biofilms under static conditions. The wells were rinsed twice with PBS to remove planktonic bacteria, air-dried at 37 °C for 30 min and stained with 0.1% crystal violet for 30 min at room temperature [5]. Excess dye was rinsed out of the wells with running water and dried in a thermostat at 37 °C for 30 min for the removal of residual water. An amount of 200 μL of 95% ethanol was added to each well and the plate was incubated at 37 °C for 3 h to ensure that the dye attached to the biofilm was completely dissolved. According to the study of Djordjevic [39], biofilm formation was quantified at 595 nm based on OD values using an enzyme marker.

#### 4.3.2. Biofilm Structure Detection

The biofilm was obtained in accordance with the method in Section 4.3.1, and it was stained with crystal violet. The changes in biofilm structure before and after treatment with the anti-microbial peptide Citrocin were observed under a light microscope.

### 4.4. Biofilm Elimination Tests

#### 4.4.1. Biofilm Biomass Determination

Biofilm quantification was performed using crystal violet staining. Bacteria grown to a stable stage were transferred to 96-well plates and incubated at 37 °C for 72 h for the generation of mature biofilms. After rinsing twice with PBS, different concentrations of Citrocin solution were added and incubated for 4 h. The Citrocin solution was removed by washing the plates three times with PBS. Biofilm quantification was performed as described in Section 4.3.1.

#### 4.4.2. Biofilm Metabolic Activity Assay

Biofilm activity was determined using a cellular activity assay (the XTT method) [40,41]. After the processes mentioned in Section 4.4.1, 10 μL of XTT working reagent solution and 50 μL of PBS were added to each well and incubated at 37 °C for 24 h. The absorbance at 450 nm was measured using a microplate reader.

### 4.5. Plate-Based Assay for Swarming Motility

The Citrocin with the final concentration of 1/4MIC, 1/2MIC, MIC, 2MIC, and 4MIC was mixed into the semi-solid medium. After the semi-solid medium solidified, 5 μL of bacterial solution was added to the center of the medium, and incubated at 37 °C for 3 d. PBS was used as a blank control. The diameter of each colony was measured [42,43]. 

### 4.6. Determination of Extracellular Polysaccharide Content

#### 4.6.1. Preparation of Standard Curve

The phenol-sulfuric acid method was used to determine the total extracellular polysaccharide in the sample. Briefly, distilled water was added into 0.4, 0.6, 0.8, 1.0, 1.2, 1.4, 1.6 and 1.8 mL of glucose solution (0.04 mg/mL) to make a total volume of 2.0 mL, while 2.0 mL of distilled water was used as a blank control. Then, 1.0 mL of 6% phenol and 5.0 mL of concentrated sulfuric acid were added, shaken and incubated for 30 min. The OD_490_ values were measured. The standard curve was plotted using the polysaccharide mass (μg) as the horizontal coordinate and the absorbance as the vertical coordinate [44].

#### 4.6.2. Preparation of Crude Polysaccharide

After centrifugation of the bacterial solution, the supernatant was transferred into a new centrifuge tube and mixed with 3 times the volume of 95% ethanol. All the tubes were placed in the refrigerator at 4 °C for alcohol precipitation overnight, after which the precipitate was later collected via centrifugation at 10,000 r/min for 10 min and re-solubilized. Then, 1.0 mL of distilled water, 1.0 mL of 6% phenol and 5.0 mL of concentrated sulfuric acid were added to 1.0 mL of the extracellular polysaccharide sample. The mixture was shaken on a vortexer and incubated for 30 min. The OD_490_ value was measured and brought into the standard curve to obtain the polysaccharide concentration [45].

### 4.7. Metabolite Analysis

#### 4.7.1. Collection of Biofilm

Activated *P. aeruginosa* was inoculated in the LB medium to a concentration of 10^7^ CFU/mL. The bacterial solution was added to cell culture plates (30 mL per well) and incubated at 37 °C for 48 h. Biofilms were collected with a cell spatula and transferred to 2 mL centrifugal tubes, ensuring that the wet weight of each sample was 100 mg. Citrocin solution at concentrations of 0.15 and 0.3 mg/mL were added onto the collected biofilms for 4 h. After centrifugation at 4 °C, the supernatant was removed, and the bacterial precipitate was collected and stored at −20 °C. An equal amount of PBS-treated biofilm was used as a negative control.

#### 4.7.2. Extraction of Metabolites

Each sample was accurately weighed to 50 mg, and dissolved in 400 μL of 80% (*v*/*v*) methanol aqueous solution. Bacterial cells were crushed at −20 °C, vortexed and shaken for 30 s, and extracted via ultrasonication at 40 kHz for 30 min at 5 °C, and then the samples were kept at −20 °C for 30 min. After centrifugation at 3000× *g* for 15 min at 4 °C, the supernatants were transferred to LC-MS/MS vials. A QC sample consisting of a mixture of all assayed samples was inserted in every 3 analyzed samples. 

#### 4.7.3. LC-MS/MS Analysis

An ultra-high-performance liquid chromatograph (UHPLC) (Ultimate 3000, Dionex, Sunnyvale, CA, USA) equipped with a Hypersil GOLD C18 column (2.1 × 100 mm, particle size 1.9 μm, Thermo Fisher Scientific, Waltham, MA, USA) was used for LC-MS/MS analysis. The flow rate was set at 0.2 mL/min, the column temperature was 35 °C and the injection volume was 2 μL. The mobile phases consisted of solvent A (ultrapure water containing 0.1% formic acid), solvent B (methanol solution containing 0.1% formic acid), solvent C (ultrapure water containing 0.1% NH_3_) and solvent D (methanol solution containing 0.1% NH_3_). The gradient program for positive- and negative-ion-mode elution is shown in Table 1. MS/MS spectra were acquired in information-dependent acquisition (IDA) mode using Q ExactiveOrbitrap under the control of Xcalibur 4.2 (Thermo Fisher Scientific, USA). The HESI (heated electrospray ionization) source conditions were adjusted as follows: the sheath and auxiliary gas flow rates were 40 and 10 Arb, respectively, capillary temperature was 320 °C, full mass scan (*m*/*z* 70–1050) resolution was 70,000, MS/MS scan mode was set to a data-dependent ms^2^ (dd-ms^2^) scan with a resolution of 35,000, and high collisional dissociation was 20/40/60 eV [46]. The spray voltage was 3.5 kV (+)/−2.5 kV (−).

### 4.8. Data Analysis

Statistical analysis was conducted through Microsoft Excel 2016 and Graphpad Prism 8.0.2 software, and an unpaired t-test was used to test the significance of differences. Raw data were preprocessed using Compound Discoverer 2.1 to obtain matched and aligned peak data, and then peaks were compared with data in the mzCloud and mzVault databases for qualitative and quantitative results. The preprocessed data were imported into SIMCA-P 14.1 for principal component analysis (PCA) and orthogonal partial least squares discriminant analysis (OPLS-DA). Potential differential metabolites were screened based on variable importance in projection (VIP) ≥ 1 and *p* < 0.05. The data were visualized with Metaboanalyst 5.0 (https://www.metaboanalyst.ca accessed on 20 June 2023). Metabolic pathways were constructed according to the KEGG metabolic database via pathway analysis on Metaboanalyst. All experiments were repeated three times, and *p* < 0.05 and *p* < 0.01 were considered significant (*) and highly significant (**), respectively.

## 5. Conclusions

Citrocin can inhibit early biofilm formation and eradicate mature biofilms. It can also reduce the motility of biofilm cells by accelerating amino acid metabolism and polysaccharide metabolism and decomposing the EPS of biofilm, thus showing a certain therapeutic effect on biofilm formation and removal in *P. aeruginosa*. In short, Citrocin, an anti-microbial peptide, is a highly effective anti-microbial agent. Further exploration of the anti-biofilm mechanism of Citrocin will contribute to the understanding of *P. aeruginosa* pathogenesis and the treatment of chronic diseases associated with biofilm-forming pathogens.

## Figures and Tables

**Figure 1 ijms-25-04122-f001:**
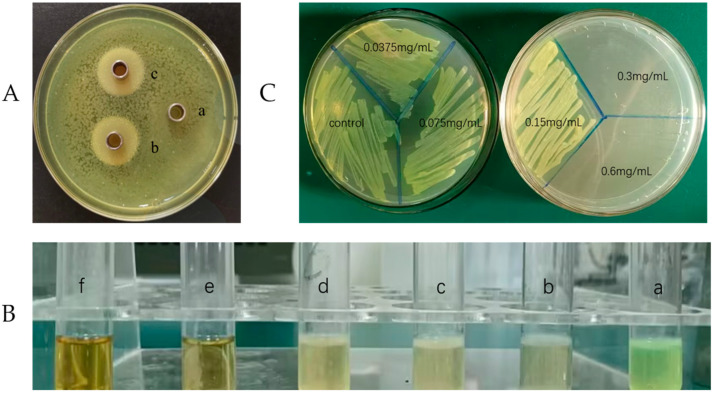
(**A**) The effect of the anti-microbial peptide Citrocin against *Pseudomonas aeruginosa* (a: control; b: 0.3 mg/mL Citrocin; c: 0.3 mg/mL Citrocin after heating at 121 °C); (**B**) the MIC of Citrocin against *P. aeruginosa* (a: control; b: 0.0375 mg/mL; c: 0.075 mg/mL; d: 0.15 mg/mL; e: 0.3 mg/mL; f: 0.6 mg/mL); (**C**) the MBC of Citrocin against *P. aeruginosa*.

**Figure 2 ijms-25-04122-f002:**
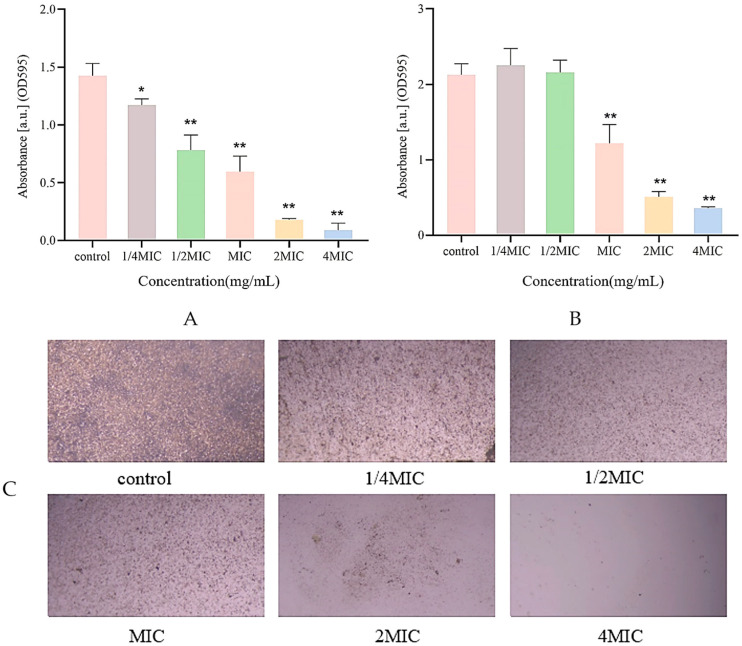
(**A**) The effect of different concentrations of Citrocin on biofilm formation; (**B**) the effect of different concentrations of Citrocin on the biomass of the established biofilm of *P. aeruginosa*; (**C**) the effect of the anti-microbial peptide Citrocin on the structure of *P. aeruginosa* biofilm. (*) represents *p* < 0.05; (**) represents *p* < 0.01.

**Figure 3 ijms-25-04122-f003:**
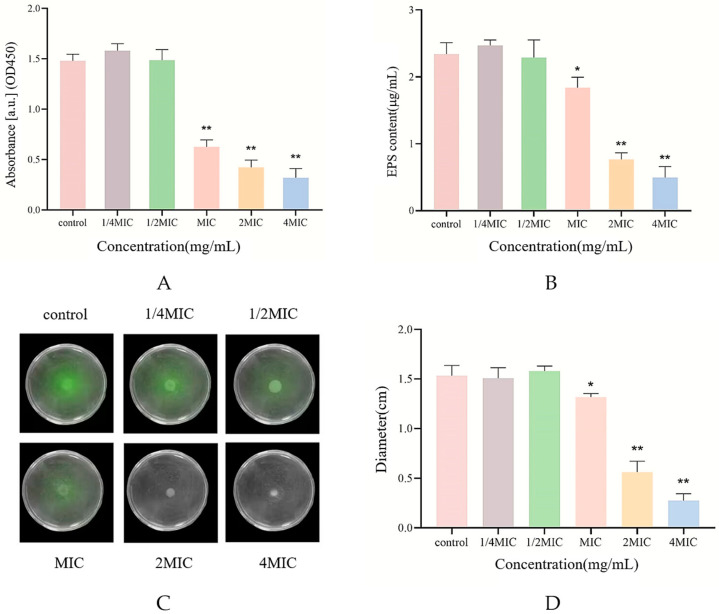
(**A**) The effect of different concentrations of Citrocin on metabolic activity of biofilm; (**B**) the effect of different concentrations of Citrocin on extracellular polysaccharide production; (**C**) the effect of swarming movement; (**D**) the effect of different concentrations of Citrocin on the swarming movement of *P. aeruginosa*. (*) represents *p* < 0.05; (**) represents *p* < 0.01.

**Figure 4 ijms-25-04122-f004:**
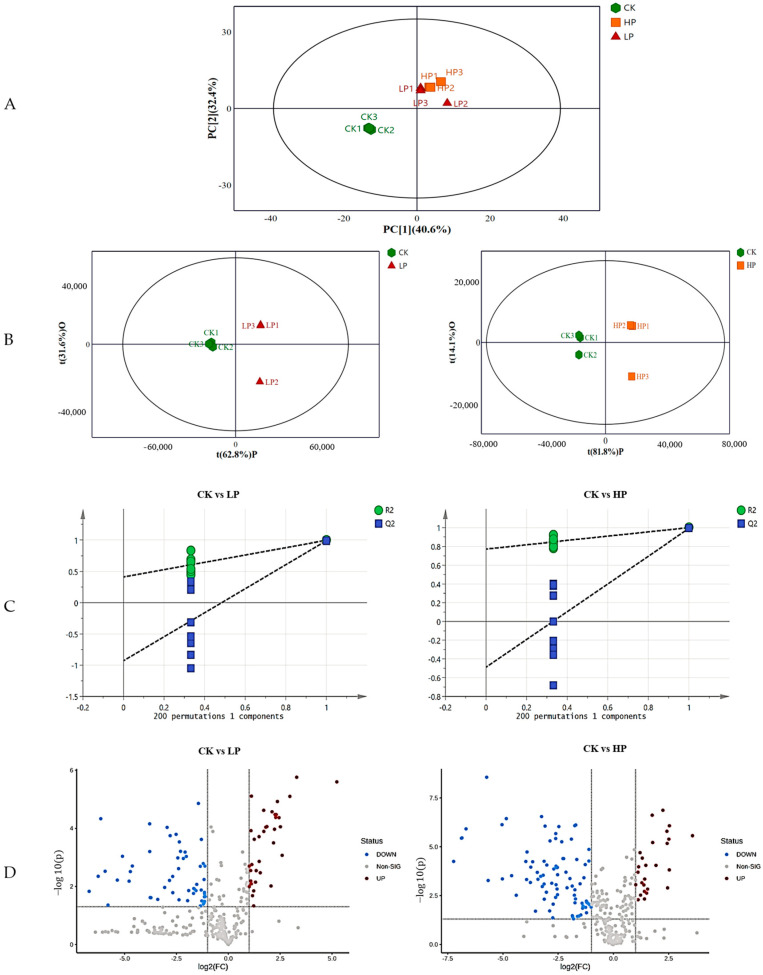
(**A**) PCA score plot of *P. aeruginosa* in response to different treatments. PC [1]: the first principal component score; PC [2]: the orthogonal principal component score. (**B**) OPLS−DA plots for different treatments. tP: the first principal component score; tO: the orthogonal principal component score. (**C**) OPLS−DA permutation plots of the validated models; (**D**) volcano plot of the differential metabolites in response to different treatments. CK: control; LP: 0.15 mg/mL Citrocin; HP: 0.3 mg/mL Citrocin.

**Figure 5 ijms-25-04122-f005:**
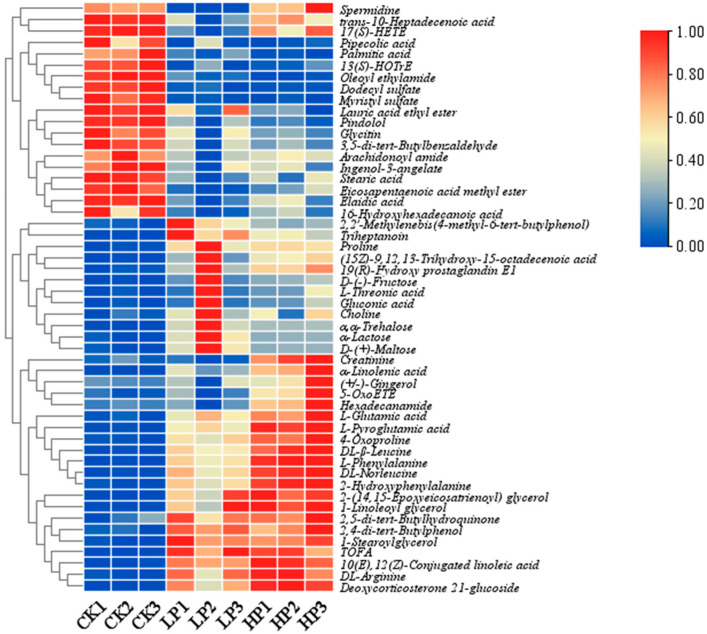
Heatmap analysis for the identified metabolites at different Citrocin concentrations (OPLS-DA (VIP > 1) and *p* < 0.05). Blue and red colors represent low and high abundance, respectively. CK: control; LP: 0.15 mg/mL Citrocin; HP: 0.3 mg/mL Citrocin.

**Figure 6 ijms-25-04122-f006:**
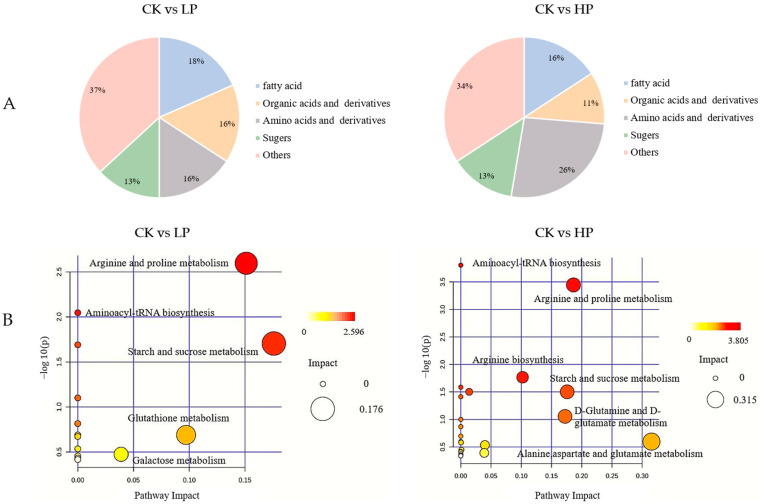
(**A**) Classification of metabolites satisfying VIP (OPLS-DA) > 1 and *p* < 0.05; (**B**) KEGG pathway analysis of differential metabolites in *P. aeruginosa* biofilm under different treatments.

**Figure 7 ijms-25-04122-f007:**
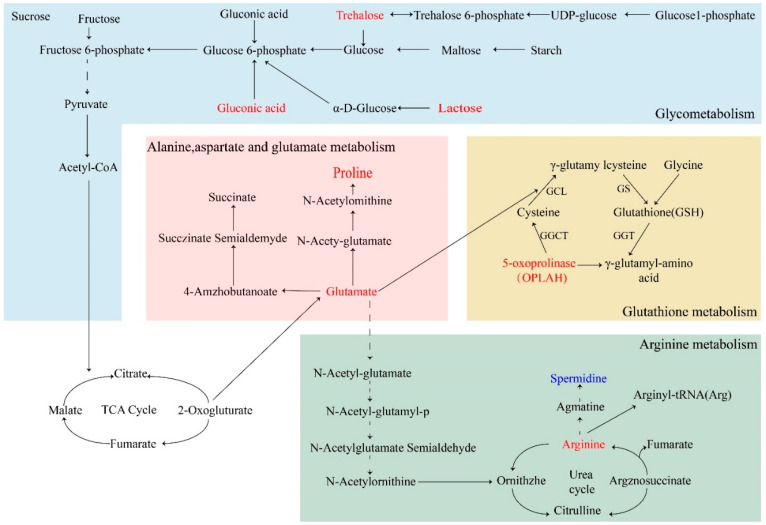
Metabolic pathway network diagram. Blue color represents down-regulation and red color represents up-regulation.

**Table 1 ijms-25-04122-t001:** Gradient program for positive- and negative-ion-mode elution.

Paradigm	Time	Ingredient
E+	0–10 min	5% A and 95% A
	10–12 min	5% A and 95% B
	12–13 min	5% A and 95% B
	13–14 min	95% A and 5% B
E−	0–2.5 min	95% C and 5% D
	2.5–16.5 min	95% D and 5% C
	16.5–19 min	95% D and 5% C
	19–20 min	95% C and 5% D

## Data Availability

Data are contained within the article.

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
