# Peer review of "The Anti-Microbial Peptide Citrocin Controls Pseudomonas aeruginosa Biofilms by Breaking Down Extracellular Polysaccharide"

_ijms, 2024, doi:10.3390/ijms25074122_

Round 1

Reviewer 1 Report

Comments and Suggestions for Authors

In the manuscript entitled "The antimicrobial peptide Citrocin controls Pseudomonas aeruginosa biofilms by breaking down extracellular polymers", a study is presented evaluating the antimicrobial and anti-biofilm properties of the Citrocin peptide, as well as the mechanism of action and its effects on the biofilm of the Pseudomonas aeruginosa bacterium.

I appreciate the novelty of this study, however, recommend the publication of this manuscript after a minor revision.

I suggest the following to the authors:

1. Correcting the numbers of the figures and correlating them correctly with the figures in the entire manuscript.

2. Replacing figures 3, 5 and 7 with ones with a better resolution.

3. Arrangement of citations in the text and in the References chapter according to the journal's requirements.

4. The antimicrobial and anti-biofilm performance of the Citrocin peptide could be compared with the activity of other peptides on the Pseudomonas aeruginosa bacteria (possibly a table).

Reviewer 2 Report

Comments and Suggestions for Authors

The manuscript titled "The antimicrobial peptide Citrocin controls Pseudomonas aeruginosa biofilms by breaking down extracellular polymers" by Liyao Wang focuses on an interesting topic. This study was conducted at the Grassland Microbiology Laboratory of Southwest University on the antimicrobial peptide citrocin and its inhibitory role on Pseudomonas aeruginosabiofilms. Congratulation for this work was a pleasure in assessing it. 

Based on the content reviewed in the manuscript, there are few examples of areas for potential improvement in English usage.

1. Spelling and Typographical Errors.

Ensure that all medical terms, names of tests, and technical languages are spelled correctly. Any typographical errors such as missing letters or incorrect punctuation should be corrected.

2. Grammar and Syntax:

Subject-Verb Agreement: Ensure that subjects and verbs in sentences agree in numbers (singular or plural). 

3. Consistency:

Ensure consistency in terminology use. Consistent formatting of references and citations according to the journal guidelines.

4. Clarity and Conciseness.

Simplifying complex sentences to enhance readability.

5. Scientific Formatting:

Verify that figures, tables, and references are correctly cited in the text and that their formatting aligns with the publication standards.

These suggestions were aimed at refining the manuscript to meet high academic standards in terms of language and presentation.

The main question addressed by the research is clear and relevant, and the topic is original, adding valuable insights to the subject area. The paper is well written, with clear and easy-to-read text that would definitely benefit from the suggestion provided in my review, so that the overall quality would be even more academic. These conclusions are consistent with the evidence and arguments presented and effectively address the main question posed.

The action of citrocin against Pseudomonas aeruginosa biofilms offers a potential therapeutic avenue, as it can disrupt the biofilm structure or inhibit its formation, thereby enhancing the efficacy of conventional treatments and reducing the resistance of bacteria. Hopefully, we will also see data from clinical trials.

Reviewer 3 Report

Comments and Suggestions for Authors

Dear Authors, 

the topic of your research is very interesting but I have a few comments on the way the results are presented: 

1. In "Abstract" you mentioned about using citrocin in animal feed. But there is no further information about this in the introduction - about the concentration used or safe of use. Also, the discussion lacks a comparison of the concentrations used in your studies to those used in feed. Information about posibble use for humans/animals and cytotoxicity of citrocin or its safety of use would be desireble.

2. There is no information about number of repetitions in each method

3. I can see some inconsistency in typing names of sugars, proteins, peptides and other words (Sub-MIC). You capitalize first letter sometimes, sometimes not. According to my knowledge there is no need to use capital letters in these words.

4. It is hard to read "Results" when the text describing figure is far away from this figure. In my opinion you should expand results descriptions (especially first 8 points concerning classic microbiological methods. Metabolomic tests are better described). Bigger charts next to its description in main text could improve clarity of this section. 

5. The numbering of the figures is incorrect in relation to the text. It is probably caused by adding graphical abstract as a Figure 1 at the end of editing process. You shoud correct this. 

6. Please add wavelenght and units to axes labels on your charts.

7. Heat map (incorrectly marked as the Fig. 6.) is very illegible. Please change the size of letters or do not bold text. What means 1, 2, 3 in CK, LP and HP? Repetitions or concentrations? If concentrations - which ones among tested?

Round 2

Reviewer 3 Report

Comments and Suggestions for Authors

Dear Authors, thank you for making the corrections.